# Posterior Fossa Tumours in the First Year of Life: A Two-Centre Retrospective Study

**DOI:** 10.3390/diagnostics12030635

**Published:** 2022-03-04

**Authors:** Stefania Picariello, Pietro Spennato, Jonathan Roth, Nir Shimony, Alessandra Marini, Lucia De Martino, Giancarlo Nicosia, Giuseppe Mirone, Maria Serena De Santi, Fabio Savoia, Maria Elena Errico, Lucia Quaglietta, Shlomi Costantini, Giuseppe Cinalli

**Affiliations:** 1Neuro-Oncology Unit, Department of Paediatric Oncology, Santobono-Pausilipon Children’s Hospital, Via Mario Fiore 6, 80129 Naples, Italy; s.picariello@santobonopausilipon.it (S.P.); l.demartino1@santobonopausilipon.it (L.D.M.); 2Department of Women, Child and General and Specialized Surgery, University of Campania “Luigi Vanvitelli”, Via Luigi De Crecchio 4, 80138 Naples, Italy; 3Division of Neurosurgery, Department of Neurosciences, Santobono-Pausilipon Children’s Hospital, Via Mario Fiore 6, 80129 Naples, Italy; pierospen@gmail.com (P.S.); peppemirone@hotmail.com (G.M.); serena.desanti@tiscali.it (M.S.D.S.); giuseppe.cinalli@gmail.com (G.C.); 4Departments of Pediatric Neurosurgery, Dana Children’s Hospital, Tel Aviv Medical Center, Tel Aviv University, 6 Weizmann St., Tel Aviv 6423906, Israel; jonaroth@gmail.com (J.R.); nirsh8@gmail.com (N.S.); sconsts@netvision.net (S.C.); 5Department of Neurosurgery, Ospedale Santa Maria della Misericordia di Perugia, Piazzale Giorgio Menghini 3, 06129 Perugia, Italy; marini.alessandra.am@gmail.com; 6Neurosurgery Unit, San Pio Hospital, Via Pacevecchia 53, 82100 Benevento, Italy; giancarlo.nicosia.gn@gmail.com; 7Childhood Cancer Registry of Campania, Santobono-Pausilipon Children’s Hospital, Via della Croce Rossa 8, 80129 Naples, Italy; f.savoia@santobonopausilipon.it; 8Department of Pathology, Santobono-Pausilipon Children’s Hospital, Via Posillipo 226, 80123 Naples, Italy; mariaelenaerrico@virgilio.it

**Keywords:** infratentorial tumours, posterior fossa tumours, infants, Atypical Teratoid/Rhabdoid Tumour, medulloblastoma, astrocytoma, ependymoma

## Abstract

Posterior fossa tumours (PFTs) in infants are very rare, and information on these tumours is scarce in the literature. This retrospective study reports their pathological characteristics and describes surgical aspects and treatment outcomes. A two-centre cohort of infants with PFTs treated from 2007 to 2018 was retrospectively reviewed. Patient characteristics, clinical, and treatment data were reviewed. Survival curves for progression-free survival (PFS) and overall survival (OS) were generated. Thirty-three infants were retrieved. There were 11 low grade and 22 high-grade tumours. The most common presenting symptom was intracranial hypertension. Fifteen children out of thirty-three progressed. Five-year PFS was significantly lower in children with high-grade tumours (38.3%) than those with low-grade tumours (69.3%), *p* = 0.030. High-grade pathology was the only predictor of progression (HR 3.7, 95% CI 1.1–13.31), *p* = 0.045. Fourteen children with high-grade tumours died, with a 5-year OS of 55.25%. PFTs in children below one year of age still represent a unique challenge. Infants with high-grade tumours display the worst outcomes and the lowest survival, indicating that more effective strategies are needed.

## 1. Introduction

Primary CNS (Central Nervous System) tumours are the most common solid tumours in children, ranging between 16% and 23% of all paediatric malignancies [1,2,3,4,5].

Brain tumours can appear anywhere across the infra and supratentorial compartments, with the posterior fossa being the most common location [3]. In children older than one year, over two-thirds of intracranial tumours arise from the cerebellum or brainstem [6,7]. Conversely, the infratentorial location is very rare in children below one year of age; hence relevant aspects of clinical, therapeutic, and biological characterisation are still undefined [8,9,10,11].

Signs of raised intracranial pressure, such as macrocephaly, bulging fontanel, and vomit, are the most common onset manifestations, though infants can present with more subtle clinical features, including hypotonia, irritability, poor feeding, failure to thrive, drowsiness, abnormal eye movement, lethargy, and a delay in developmental milestones [9,12]. Additional signs and symptoms might be found depending on the tumour location and involvement of anatomical structures and cranial nerves [13,14].

Posterior fossa brain tumours (PFTs) in infancy include a wide range of histopathological types with different biological behaviours and clinical management. Despite the advances in research, providing a more thorough understanding of molecular profiling alongside diagnostic and treatment modalities, the outcome of infants with primary CNS tumours is still poor [15,16,17,18].

Surgical resection can be difficult in very young children because of the increased risk of complications—hypotension, hypoxia, hypothermia, and bleeding, associated with the persistence of foetal dural venous circulation pattern and high-grade hypervascular tumours. Not least, the impaired and delayed wound healing together with adjuvant therapy toxicity can greatly contribute to additional morbidity [19,20].

In this 10-year retrospective study, we aimed to provide a description of the clinical and pathologic characteristics of this characteristics of PFTs and to report our experience on management and survival outcomes of infants with posterior fossa tumours.

## 2. Materials and Methods

This is a two-centre retrospective study of infants diagnosed with primitive posterior fossa tumours at the Department of Paediatric Neurosurgery, Santobono-Pausilipon Children’s Hospital, Naples, Italy and Dana Children’s Hospital, Tel-Aviv Medical Center, Tel Aviv, Israel, between January 2007 and January 2018.

Patients were identified through a review of electronic clinical datasets. Only children with pathology-confirmed tumour diagnosis and available pre and postoperative MRI were included. As per standard clinical practice, all cases were discussed at Neuro-oncology multidisciplinary meetings in both Hospitals.

Demographic and relevant data (presenting symptoms, tumour pathology, surgical management, treatment modalities, tumour recurrence/progression, cause of death) were recorded.

Surgical management of hydrocephalus was classified as external ventricular drainage (EVD), Endoscopic Third Ventriculostomy (ETV), ventriculoperitoneal shunt (VP shunt).

Regarding the surgical approach to the tumour, a prone position with a midline skin incision was adopted in most of the procedures, allowing wide access to the posterior fossa. However, for tumours extending to the cerebellopontine angle, the lateral or park bench position with lateral suboccipital craniotomy was preferred. All patients were transferred post-operatively in the paediatric intensive care unit (PICU) for observation and stabilisation.

Brain MRI with contrast medium administration within 24 h from surgery was performed to rule out neurosurgical complications and to assess the extent of resection: gross total resection (GTR) (no residual disease), subtotal resection (STR) (90% reduction in tumour size), partial resection (50–90% reduction in tumour size), and biopsy (<50% reduction in tumour size).

Tumour pathology was assigned according to the 2007 and 2016 World Health Organisation Classification of the Central Nervous System, and tumours were grouped into a low grade (0, I and II) and high grade (III and IV) CNS tumours. Metastatic diseases were staged as M0: non metastasis, M1: cells in cerebrospinal fluid (CSF), M2: cranial metastasis, M3: spinal metastasis, and M4: extraneural metastasis.

### Statistical Analysis

Continuous nonparametric variables are reported as median (range), categorical variables as number and percentage. Differences between groups were compared by Chi-square or Fisher’s exact tests for categorical variables and the Mann–Whitney *U* test for continuous nonparametric variables.

Kaplan–Meier method was used to estimate survival curves for progression-free survival and overall survival. Progression-free survival (PFS) was defined as the time from diagnosis to the first progression, recurrence, or metastatic dissemination. Overall survival (OS) was defined from the diagnosis until death. Patients without events were censored at last follow up or death for PFS. The log-rank test was used to compare survival curves. Cox proportional hazard model was used to estimate hazard ratio and 95% Confidence Interval (CI) for predictors of progression.

A *p* value ≤ 0.05 was considered significant. The software SPSS-IBM 22 was used for all analyses.

## 3. Results

From 2007 to 2018, 33 children under 12 months of age underwent a surgical procedure for a posterior fossa tumour in our Departments (18 at Santobono-Pausilipon Children’s Hospital, and 15 at Tel Aviv Medical Center).

The median age at diagnosis was 7.3 (1–12) months, with 15 children (45.5%) being younger than six months of age. Twenty-three out of thirty-six patients (70%) were female.

All children were symptomatic, and none of them had a prenatal diagnosis of PFT. The most common presenting symptom was intracranial hypertension (ICHT) (vomiting, headache, enlarged head circumference, rapid deterioration, sleepiness). It was documented in 27 children out of 33 (81.8%), associated with cerebellar signs (gait disturbances) in four children, torticollis in five, Parinaud’s syndrome in three cases, dysphagia in two, strabismus in two, facial nerve palsy in one.

In most cases (23/33, 69.7%), the tumour was located in the midline (cerebellar vermis, brainstem), in two cases in the cerebellar hemisphere (6%), seven in the cerebellopontine angle (21.2%), and one in the pineal region.

CNS WHO grade 0–II tumours were diagnosed in 11 children (33.3%), whilst high-grade tumours (grade III or IV) in 22 (66.7%). No difference was found in age at diagnosis between children with high-grade and those with low-grade tumours (*p* = 0.693).

Atypical Teratoid/Rhabdoid tumour (AT/RT) was the most common pathology (9, 27%), followed by pilocytic astrocytoma (7, 21.2%) and medulloblastomas (7, 21.2%). Five cases of WHO grade III ependymoma (15.1%) and one WHO grade II ependymoma were also registered, together with single cases of mature teratoma, desmoplastic infantile glioma, high-grade glioma, and pilomyxoid astrocytoma.

Metastatic disease at diagnosis was documented in three cases of desmoplastic nodular medulloblastoma (2 M2, 1 M1) and two cases of AT/RT (1 M1, 1 M3).

### 3.1. Management of Hydrocephalus

Clinical and radiological findings of hydrocephalus were witnessed in 27 patients (81.8%), 9/11 (81.8%) children with low-grade pathology and 18/22 (81.8%) with high-grade tumours (*p* = 1.000).

Surgery for hydrocephalus was carried out in 23 children prior to tumour-directed surgery. In four cases, upfront tumour resection was performed because of the relevant mass effect and neuroradiological evidence of obstructive hydrocephalus.

Ventriculoperitoneal (VP) shunt was implanted in 14 children (42.4%), EVD was initially preferred in 4 patients, associated with ETV in 2 cases. Endoscopic third ventriculostomy was performed at admission in further five patients, in association with Ommaya reservoir in two of them.

However, four out of seven children treated with ETV experienced ETV failure and persistent hydrocephalus after tumour resection and required a VP shunt placement.

### 3.2. Survival Outcome

Patients were followed for a median follow up time of 43 (1–117) months.

Fifteen children out of 33 progressed with a whole population 5-year (60 months) PFS of 49.2% (Figure 1).

PFS curves were significantly different between children with low-grade tumours (3/11 events, 5-year PFS 69.3%) and those with high-grade tumours (12/22 events, 5-year PFS 38.3%), *p* = 0.030 (Figure 2).

High-grade pathology was indeed the only predictor of progression in the univariate Cox Regression analysis (HR 3.7, 95% CI 1.1–13.31, *p* = 0.045), while the other explored variables were not found statistically significant (ICHT *p* = 0.411; age at diagnosis *p* = 0.115; GTR *p* = 0.165).

Fourteen children died after a median of 6.5 (1–44) months since diagnosis, at a median age of 13 (2–56) months, with a 5-year OS of 55.25%. All of them were affected by high-grade tumours (Figure 3).

### 3.3. Treatment Details and Outcome Stratified by WHO Grade

#### 3.3.1. Grade 0 (One Patient)

A pineal mature teratoma underwent subtotal resection and did not recur over more than five years of follow up.

#### 3.3.2. Grade I (Eight Patients)

Seven grade I tumours were pilocytic astrocytomas. GTR was obtained in five of them, STR in one, one child had brainstem invasion requiring a more conservative approach (partial removal) followed by standard Vincristine–Carboplatin chemotherapy treatment [21]. The latter progressed 24 months after diagnosis and was completely resected. Lastly, a desmoplastic infantile glioma of the brainstem was subjected to biopsy at diagnosis and underwent a watch and wait strategy with stable disease 15 months after diagnosis.

#### 3.3.3. Grade II (Two Patients)

One case of WHO grade II Ependymoma was treated as per SIOP Ependymoma I Protocol [22]. A subtotal resection was achieved, but the residual lesion progressed 12 months later, and the child was treated with GTR followed by radiotherapy. The child was alive with no evidence of disease at 35 months of follow-up.

The second child had an exophytic brainstem pilomyxoid astrocytoma and underwent subtotal resection and chemotherapy according to Vincristine–Carboplatin chemotherapy treatment [21]. Recurrence occurred 40 months later and was treated with surgery and second-line chemotherapy with no further progression at 69 months of follow-up.

#### 3.3.4. Grade III (Five Patients)

All these patients had anaplastic ependymomas. Three children were treated with GTR resection and chemotherapy, per Italian protocol for paediatric intracranial ependymoma [23]. All children relapsed during follow-up: two patients progressed at 9 and 17 months after diagnosis and died during their fourth year of follow-up. One child progressed 22 months after diagnosis and was alive with no evidence of disease at 52 months of follow-up after surgical resection of a single lumbar metastasis followed by focal radiotherapy (Figure 4).

Two children were treated per SIOP Ependymoma I Protocol [22]. The former had metastatic spinal dissemination six months after the GTR surgery and was treated with extensive surgical removal, radiotherapy, and metronomic treatment. Unfortunately, the child died after two years of follow-up because of multiple disease progressions. A GTR was initially achieved in the latter, with subsequent tumour progression at 10 months. A second surgical excision was attempted (GTR) followed by radiotherapy with a stable remnant lesion at 81 months.

#### 3.3.5. Grade IV (17 Patients)

Nine grade IV patients were affected by AT/RT, metastatic in two cases (one M1, one M3); all of them underwent a surgical procedure (two GTR, two STR, four partial resection and one biopsy) followed by adjuvant therapy based on European Rhabdoid Registry consensus therapy recommendation [24]. Two patients only survived during the follow-up.

Seven patients had a diagnosis of medulloblastoma, namely large cell medulloblastoma in one case and desmoplastic/nodular pathology in six cases, with cranial metastases in two. The former underwent a biopsy and died of gastro-intestinal haemorrhage due to acute chemotherapy toxicity two months later. The other patients had surgery (one partial removal and five GTR) and induction chemotherapy followed by high dose myeloablative chemotherapy with autologous stem cell rescue (HDC/ASCR). Two of them died of complications several months after treatment: one patient developed chronic renal and cardiac failure associated with myelodysplasia, resulting in hemocoagulative disorders. She also experienced multiple episodes of shunt failure and a mycotic ventriculitis and eventually died of massive intraventricular haemorrhage (Figure 5).

The second patient experienced cardiac failure and died when she was on mechanical circulatory support as a bridge to a heart transplant.

One child died of acute toxicity three months after diagnosis due to acute renal failure following the second cycle of chemotherapy.

The patient with high-grade glioma had a partial removal and induction chemotherapy followed by HDC/ASCR and focal radiotherapy. The child progressed two months after the end of treatment and was treated with Temozolomide and Bevacizumab, interrupted after the first cycle due to cardiotoxicity. Cyberknife radiosurgery was therefore performed. The child is now stable after six years of follow-up.

## 4. Discussion

Brain tumours in the first year of life differ in topographical distribution, tumour biology, clinical course, and prognosis compared with those diagnosed at older ages [16,25].

The prognosis is usually poor in infants, and the outcome is largely related to pathological and biological characteristics, size, and the location of the tumour [26,27,28].

This is extremely relevant to tumours involving the vital brainstem nuclei and critical cerebellar structures [29,30]. The posterior fossa is indeed the commonest site of primary brain tumours in older children, whilst the supratentorial location is more prevalent in those below one year of age, infratentorial tumours accounting for 17.9–32.4% depending on series [8,31].

In view of their rarity, posterior fossa tumours presenting in the first year of life still represent a unique challenge, and a focus of intense research aimed not just at prolonging survival but also at improving the quality of survival [11,32,33,34,35].

In this multicentre study, we focus on the clinical presentation, the histopathological diagnosis, the surgical management of hydrocephalus, treatment, and survival outcome of 33 infants with posterior fossa tumours diagnosed and followed over a decade.

Clinical and radiological findings of hydrocephalus were documented in more than 80% of children at the time of presentation in our series, in line with previous literature [9,36,37].

In our centres, CSF diversion is performed before tumour resection surgery in most infant cases, with the aim of relieving the intracranial pressure and facilitating the tumour resection. External ventricular drain placement is performed in the presence of acutely deteriorating consciousness and cases requiring prompt stabilisation, though close clinical monitoring is required afterwards to avoid overdrainage and infective complications. Alternatively, a VP shunt can be placed, particularly in cases with the metastatic or multicentric disease, whilst ETV is a valid option to temporarily control the hydrocephalus, avoiding EVD or permanent VP shunt [19,38,39,40].

To date, there is no consensus on the best management strategy of hydrocephalus in children with posterior fossa tumours, particularly in terms of the timing of the CSF diversion and the indications and efficacy of ETV and VP shunt in different clinical scenarios. The choice between the surgical technique is often dictated by the severity of symptoms and the degree of the hydrocephalus, together with the personal choice of the surgeon [36,41,42].

The majority of children with PFT-related ICHT usually experience resolution after removal of the underlying disease, whilst about 10–40% have persistent hydrocephalus. Several risk factors associated with permanent CSF diversion in children with PFT have been identified in previous studies, namely age < 2–3 years at surgery, perioperative EVD placement, postresection complications and signs of hydrocephalus in postoperative imaging, tumour type and location, metastatic dissemination, and subtotal tumour resection [43,44,45].

Some authors recommend preresection ETV in patients with a high risk of persisting postoperative hydrocephalus and in patients whose tumour surgery is delayed. In fact, despite ETV failure occurring sooner than VP shunt failure, VP shunt placement is burdened with a higher rate of postoperative complications and VP shunt revision surgeries that might delay the adjuvant treatment in the early treatment phase or dramatically affect the quality of life in the long-term survivors [38,46,47,48,49].

However, in our cohort of high-risk infants, ETV was associated with a high incidence of failure and over half of the patients eventually required VP shunt despite tumour resection. Hence, the optimal method of CSF diversion in infants with PFT remains largely debated, and hydrocephalus management requires an individualised case-based approach.

In infants with brain tumours, tumour location is correlated with the outcome, too. In fact, supratentorial tumours are usually low grade in contrast to their infratentorial counterparts, which are malignant in most cases [8,9].

In our series, the prevalence of high-grade tumours was double that of the low-grade ones. AT/RT was the most common pathology, accounting for a quarter of the study cohort. Allowing for tumour location, this particular finding is in contrast with previous literature, reporting teratoma and pilocytic astrocytoma as the most common pathology in infants, followed by medulloblastoma [6,50,51].

Besides WHO grade, we failed to identify any predictors of survival outcome. In fact, the small number of patients included in the study, together with the heterogeneity of histopathology and the variety of treatment strategies adopted, preclude any further and detailed analysis.

However, it is worthwhile to note that, despite young age being a risk factor for progression and death, none of the children with low-grade tumours died. In contrast, almost two-thirds of children with high-grade tumours experienced progression and eventually died. Among deceased subjects, over half died within six months after diagnosis, and four children affected by medulloblastoma died of treatment-related toxicity.

There is no current guideline for medulloblastoma under the age of 36 months. A trial is being developed by SIOP but has not yet opened. Multiple treatment strategies have been adopted in the treatment of very young children with posterior fossa tumours, yielding extremely variable results and still dismal prognosis, particularly for high-grade tumours [35,52,53]. 

Surgery is the mainstay of treatment for brain tumours in the first year of life, and the extent of resection is strictly related to the outcome, while radiotherapy is usually avoided in infants due to the detrimental effect on the immature and growing brain [54]. However, gross total resection is not always feasible and often encumbered by a number of immediate postsurgical complications and a higher mortality rate compared with older children [31]. High-grade pathology additionally raises the risk of postoperative complications, resulting in even higher and sobering figures [55,56]. Standard dose chemotherapy and HDC/ASCR are widely recognised as effective adjuvant treatments, particularly in cases with malignant pathology, and play an important role in delaying radiotherapy at a later stage in children with residual or recurrent disease. Despite these advantages, chemotherapy itself can be a cause of life-threatening side effects and contribute to both early and delayed mortality. Moreover, such a fragile population of young children can suffer from several late effects and long-term disability brought on by the tumour as well as the treatment strategies [57,58,59,60,61,62].

That is why advances in local surgical treatments and a better understanding of tumour molecular biology, along with the development of targeted therapies, are indispensable to provide risk stratification and to improve the now scarce chance of cure.

There are limitations to the present study that should be acknowledged owing to its retrospective design. First, beyond survival, the heterogeneity of tumours and the small numbers did not allow us to perform a detailed analysis of prognostic factors in this highly selected cohort of infants. Secondly, postoperative complications, long-term sequelae, and functional outcomes are not reported in the present study. However, the short and long-term morbidity in survivors, caused by the disease itself and the detrimental effect of multiple treatment modalities, is particularly relevant in children below one year of age and should be acknowledged [32,55,56,57,58,59,60,61,62,63].

Despite these limitations, we provide a large descriptive cohort of posterior fossa tumours presenting in infancy. These data raise awareness of the need for further research and implementation of current knowledge and clinical practice.

## 5. Conclusions

Posterior fossa tumours in children below one year old still represent a unique challenge because of their aggressive behaviour and the scarce chance of cure. The prognosis is poor and closely dependent on the histopathological grade and tumour type. Infants with high-grade tumours display the worst outcomes and the lowest overall survival, whilst those with low-grade tumours inevitably present long-term disability.

Surgery continues to be the mainstay of treatment, albeit local salvage therapy contributes to morbidity.

Efforts focused on providing advances in surgical techniques and improving molecular understanding of tumour biology are crucial in this particularly vulnerable population in order to improve the chance of cure and the long-term outcomes.

## Figures and Tables

**Figure 1 diagnostics-12-00635-f001:**
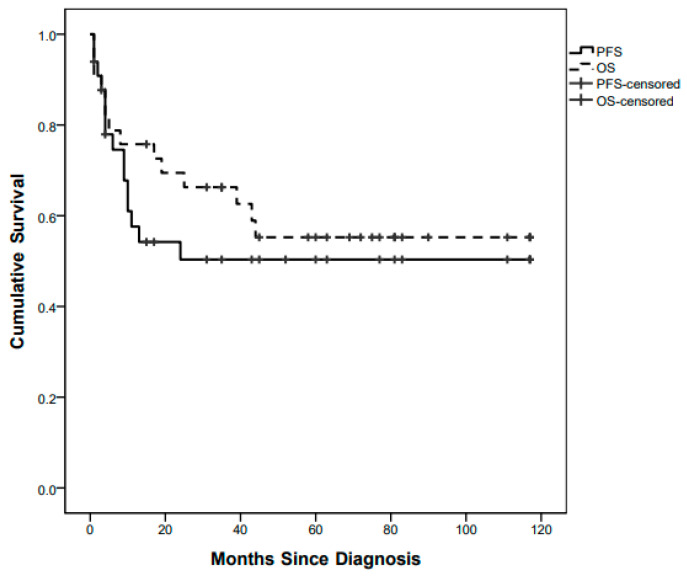
Whole-population Progression-Free Survival (PFS) and Overall Survival (OS) curves.

**Figure 2 diagnostics-12-00635-f002:**
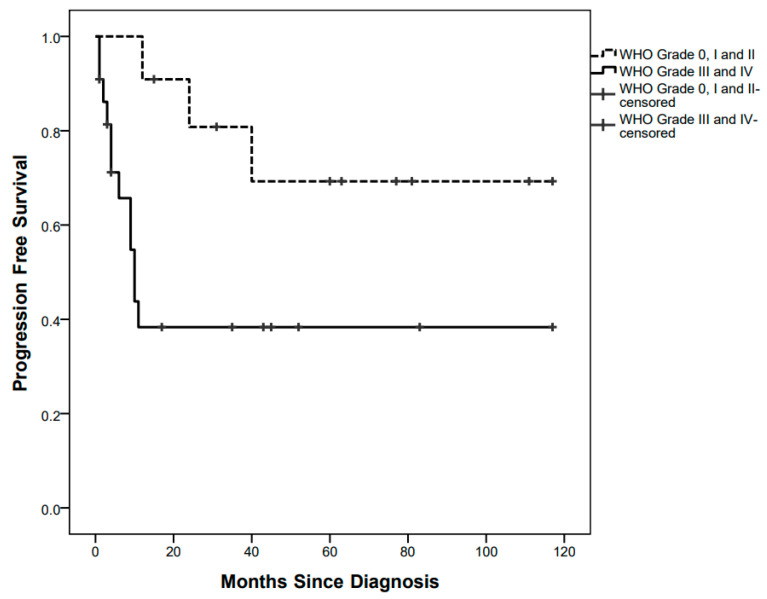
Progression-Free Survival curves by low-grade and high-grade pathology.

**Figure 3 diagnostics-12-00635-f003:**
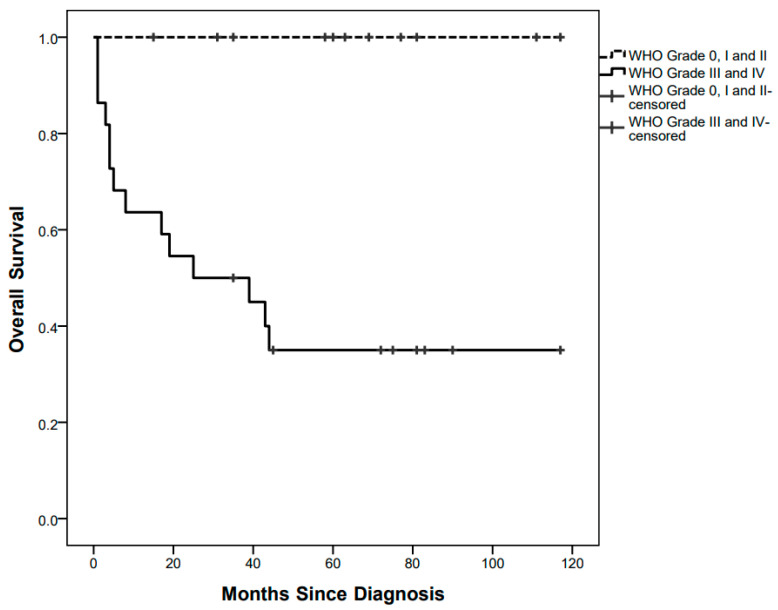
Overall survival curves by low grade and high-grade pathology.

**Figure 4 diagnostics-12-00635-f004:**
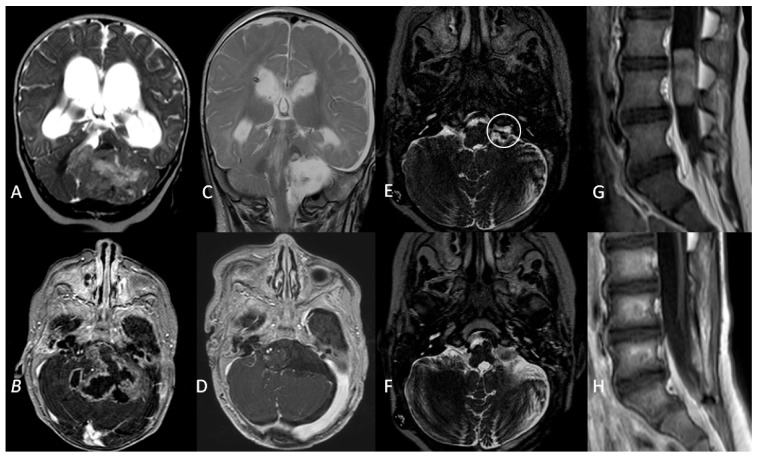
A coronal T2W MRI (**A**) and axial contrast-enhanced T1 W MRI (**B**) showing a posterior fossa tumour involving the left cerebellopontine angle and extending into the fourth ventricle in a 5-month old infant. (**C**,**D**) Three-month follow-up MRI showing complete surgical resection and VP shunt. (**E**). An axial T2W MRI at 1-year follow-up showing recurrence (white circle) of the tumour in the left lateral recess. (**F**). Gross total resection of the tumour recurrence. (**G**). Metastatic dissemination 22 months after diagnosis (single lumbar metastasis). (**H**). Complete removal of the spinal lesion.

**Figure 5 diagnostics-12-00635-f005:**
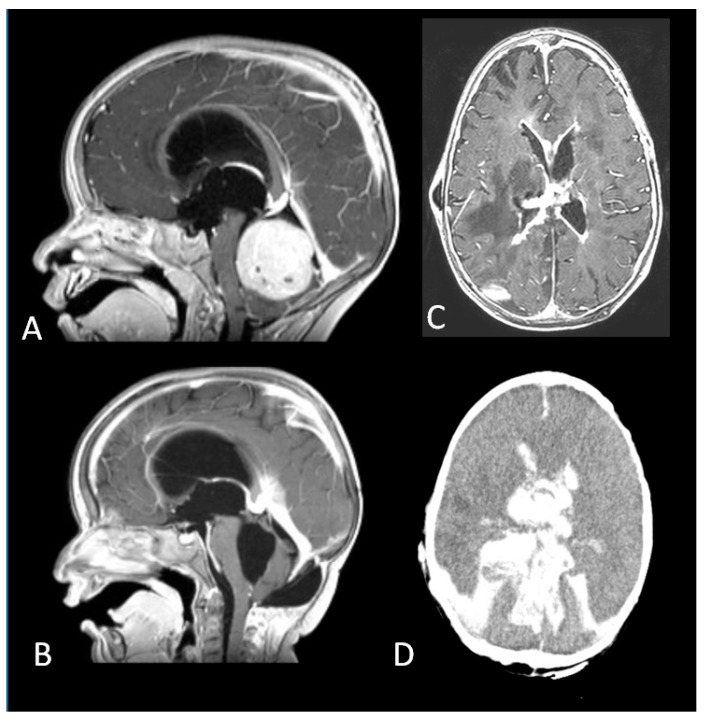
(**A**). A fourth ventricular tumour arising from the cerebellar vermis. (**B**). Complete Resection of a desmoplastic medulloblastoma. (**C**). Mycotic ventriculitis (**D**). Intraventricular haemorrhage.

## Data Availability

The data presented in this study are available on request from the corresponding author.

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
