# Peer review of "Posterior Fossa Tumours in the First Year of Life: A Two-Centre Retrospective Study"

_diagnostics, 2022, doi:10.3390/diagnostics12030635_

Round 1
Reviewer 1 Report
It's an honor for me to review the manuscript number "diagnostics-1606644" titled: "Posterior Fossa Tumours in the first year of life: a two-centre Retrospective Study" for the journal "Diagnostics".
The authors presented a very interesting original article where they report a two-centre cohort of infants with PFTs treated from 2007 to 2018.
Thirty-three infants were retrieved. There were 11 low grade and 22 high grade tumours. they found that the most common presenting symptom was intracranial hypertension, that high grade pathology was the only predictor of 35 progression (HR 3.7, 95% IC 1.1 – 13.31), p 0.045, concluding that infants with high grade tumours display the worst outcomes and the lowest survival, indicating that more effective strategies are needed.
In general, the manuscript is interesting, the investigation is acceptable, the paper is clear and well presented.
I would just suggest to better discuss clinical and radiological complications regarding posterior fossa tumours in relation to their grade (eg: Tartaglione T, et al. MRI findings of olivary degeneration after surgery for posterior fossa tumours in children: incidence, time course and correlation with tumour grading. Radiol Med. 2015 May;120(5):474-82. doi: 10.1007/s11547-014-0477-x. Epub 2015 Jan 9. PMID: 25572537.)
Author Response
Thank you for your comment and for your suggestion.
Unfortunately, clinical and radiological complications after surgery were not recorded. However, we agree with the reviewer that this is a very interesting topic that contributes to immediate and long-term morbidity. We added the following sentence to the discussion: “High grade pathology additionally raise the risk of postoperative complications, resulting in even higher and sobering figures”.
The following period has been amended too (bold text): "There are limitations to the present study that should be acknowledged owing to its retrospective design. First, beyond survival, the heterogeneity of tumours and the small numbers did not allow us to perform a detailed analysis of prognostic factors in this highly selected cohort of infants. Secondly, postoperative complications, long-term sequelae and functional outcomes are not reported in the present study. However, the short- and long term morbidity in survivors, caused by the disease itself and the detrimental effect of multiple treatment modalities, is particularly relevant in children below one year of age and should be acknowledged".
References (55-63) have been added and modified accordingly.
Reviewer 2 Report
Dear Authors,
Thank you for presenting your data on this very interesting and not so much investigated subject.
I would like to ask one question: in any of the 33 cases was a prenatal diagnosis made? I wonder if a third trimester pregnancy scan might pick up some of the cases?
Thank you
Author Response
Thank you for your opinion and for raising this point.
None of the cases had a prenatal diagnosis of PFT, and all of the children included in our study presented with intracranial hypertension and/or neurological signs.
The following sentence has been added to the result section: "All children were symptomatic and none of them had a prenatal diagnosis of PFT".